# Vitamin D levels and risk of type 1 diabetes: A Mendelian randomization study

**Despoina Manousaki**[1,2]*, **Adil Harroud**[3,4], **Ruth E. Mitchell**[5,6], **Stephanie Ross**[7], **Vince Forgetta**[8], **Nicholas J. Timpson**[5], **George Davey Smith**[5,6], **Constantin Polychronakos**[9,10,11], **J Brent Richards**[8,9,12,13,14]

**1** Department of Pediatrics, Faculty of Medicine, University of Montreal, Montreal, Quebec, Canada, **2** Research Center of the Sainte-Justine University Hospital, Montreal, Quebec, Canada, **3** Department of Neurology, University of California San Francisco, San Francisco, California, United States of America, **4** Weill Institute for Neurosciences, University of California San Francisco, San Francisco, California, United States of America, **5** Medical Research Council Integrative Epidemiology Unit, University of Bristol, Bristol, United Kingdom, **6** Population Health Sciences, Bristol Medical School, University of Bristol, Bristol, United Kingdom, **7** Department of Health Research Methods, Evidence, and Impact, McMaster University, Hamilton, Ontario, Canada, **8** Centre for Clinical Epidemiology, Department of Epidemiology, Lady Davis Institute for Medical Research, Jewish General Hospital, McGill University, Montreal, Quebec, Canada, **9** Department of Human Genetics, McGill University, Montreal, Quebec, Canada, **10** Department of Pediatrics, McGill University, Montreal, Quebec, Canada, **11** Centre of Excellence in Translational Immunology (CETI), Montréal, Quebec, Canada, **12** Department of Medicine, McGill University, Montreal, Quebec, Canada, **13** Epidemiology and Biostatistics, McGill University, Montreal, Quebec, Canada, **14** Department of Twin Research and Genetic Epidemiology, King's College London, United Kingdom

* Despina.manousaki@umontreal.ca

**Data Availability Statement:** All relevant data are within the manuscript and its Supporting Information files.

## Abstract

### Background

Vitamin D deficiency has been associated with type 1 diabetes in observational studies, but evidence from randomized controlled trials (RCTs) is lacking. The aim of this study was to test whether genetically decreased vitamin D levels are causally associated with type 1 diabetes using Mendelian randomization (MR).

### Methods and findings

For our two-sample MR study, we selected as instruments single nucleotide polymorphisms (SNPs) that are strongly associated with 25-hydroxyvitamin D (25OHD) levels in a large vitamin D genome-wide association study (GWAS) on 443,734 Europeans and obtained their corresponding effect estimates on type 1 diabetes risk from a large meta-analysis of 12 type 1 diabetes GWAS studies (Ntot = 24,063, 9,358 cases, and 15,705 controls). In addition to the main analysis using inverse variance weighted MR, we applied 3 additional methods to control for pleiotropy (MR-Egger, weighted median, and mode-based estimate) and compared the respective MR estimates. We also undertook sensitivity analyses excluding SNPs with potential pleiotropic effects. We identified 69 lead independent common SNPs to be genome-wide significant for 25OHD, explaining 3.1% of the variance in 25OHD levels. MR analyses suggested that a 1 standard deviation (SD) decrease in standardized natural log-transformed 25OHD (corresponding to a 29-nmol/l change in 25OHD levels in vitamin D–insufficient individuals) was not associated with an increase in type 1 diabetes risk (inverse-

**Funding:** DM is supported by JDRF (JDRF 3-PDF-2017-370-A-N). JBR is supported by the Canadian Institute of Health Research and Fonds de la recherche en santé du Quebec (FRSQ). A.H. is funded by the NMSS-ABF Clinician Scientist Development Award from the National Multiple Sclerosis Society (NMSS) and the Multiple Sclerosis Society of Canada (MSSC). NJT is a Wellcome Trust Investigator (202802/Z/16/Z), is the PI of the Avon Longitudinal Study of Parents and Children (MRC & WT 217065/Z/19/Z), is supported by the NIHR Biomedical Research Centre at University Hospitals Bristol and Weston NHS Foundation Trust and the University of Bristol (BRC-1215-2001), the MRC Integrative Epidemiology Unit (MC_UU_00011) and works within the CRUK Integrative Cancer Epidemiology Programme (C18281/A19169).GDS works in the Medical Research Council Integrative Epidemiology Unit at the University of Bristol MC_UU_00011/1. Funders had no role in study design, data collection and analysis, decision to publish, or preparation of the manuscript.

**Competing interests:** I have read the journal's policy and the authors of this manuscript have the following competing interests:JBR has worked as a consultant to GlaxoSmithKline and Deerfield Capital.GDS is a member of the Editorial Board of PLOS Medicine.

**Abbreviations:** 25OHD, 25-hydroxyvitamin D; CI, confidence interval; GWAS, genome-wide association study; IVW, inverse-variance weighted; LD, linkage disequilibrium; MAF, minor allele frequency; MBE, mode-based estimate; MR, Mendelian randomization; MR-PRESSO, MR pleiotropy residual sum and outlier; OR, odds ratio; RCT, randomized controlled trial; RR, relative risk; SD, standard deviation; SNP, single nucleotide polymorphism; STROBE, STrengthening the Reporting of OBservational studies in Epidemiology.

variance weighted (IVW) MR odds ratio (OR) = 1.09, 95% CI: 0.86 to 1.40, $p = 0.48$). We obtained similar results using the 3 pleiotropy robust MR methods and in sensitivity analyses excluding SNPs associated with serum lipid levels, body composition, blood traits, and type 2 diabetes. Our findings indicate that decreased vitamin D levels did not have a substantial impact on risk of type 1 diabetes in the populations studied. Study limitations include an inability to exclude the existence of smaller associations and a lack of evidence from non-European populations.

## Conclusions

Our findings suggest that 25OHD levels are unlikely to have a large effect on risk of type 1 diabetes, but larger MR studies or RCTs are needed to investigate small effects.

## Author summary

### Why was this study done?

- Observational epidemiological studies have associated low vitamin D levels with risk of type 1 diabetes; however, these studies are susceptible to confounding and reverse causation, and thus it remains unclear whether these associations are accurate.

- To our knowledge, there are no randomized controlled trials published to date on this topic.

- If vitamin D insufficiency did cause type 1 diabetes, this would be of clinical relevance for type 1 diabetes prevention in high-risk individuals, since vitamin D insufficiency is common and safely correctable.

### What did researchers do and find?

- We applied a Mendelian randomization study design to understand if vitamin D levels are associated with a higher risk of type 1 diabetes. This approach offers an alternative analytical technique able to reduce conventional patterns of confounding and reverse causation and reestimate observations in a framework allowing causal inference.

- Our study did not find evidence in support of a large effect of vitamin D levels on type 1 diabetes. However, the findings do not exclude the possibility that there may be smaller effects than we could not detect.

### What do these findings mean?

- Our findings suggest that the previous epidemiological associations between vitamin D and type 1 diabetes could be due to confounding factors, such as latitude and exposure to sunlight.

- Our results do not support increasing vitamin D levels as a strategy to decrease the risk of type 1 diabetes.

## Introduction

Type 1 diabetes is a relatively common autoimmune disease affecting the pancreatic beta cell. Its incidence is increasing worldwide [1], and it inflicts substantial life-long morbidity, affecting patients during childhood and throughout their adult life. Currently, there are no known therapies that can be used in the prevention of type 1 diabetes, but evidence exists that low vitamin D level may play a role in predisposition to this disease.

Animal studies [2–5] and observational studies [6–10] have shown that reduced levels of serum 25 hydroxyvitamin D (25OHD) are associated with an increased risk of type 1 diabetes. Additionally, it has been shown that vitamin D could prevent cytokine-induced apoptosis of human pancreatic islets [11]. In an observational birth cohort of 10,366 children, vitamin D supplementation was associated with a decreased risk of type 1 diabetes as compared to those who did not receive supplementation (relative risk (RR): 0.12, 95% CI 0.03 to 0.51) [6]. In a large prospective study on 8,676 children from the TEDDY cohort, the authors reported a weak association between 25OHD levels and odds of type 1 diabetes (odds ratio (OR) = 0.93 for a 5-nmol/L difference; 95% CI 0.89, 0.97) [12]. The most recent observational evidence comes from a nested case-control study in the TRIGR cohort, which showed a protective role of higher 25OHD levels [13].

Conversely, a recent study on 1,316 children with diagnosed type 1 diabetes showed an unexpected inverse association between vitamin D intake and fasting C-peptide, a marker of residual insulin production from the pancreatic beta cell [7]. A small prospective study in 252 Finnish children showed no difference in 25OHD levels among these who progressed to type 1 diabetes compared to controls [14]. A larger observational study in children at risk for type 1 diabetes showed that neither increased 25OHD levels nor vitamin D supplementation were associated with islet autoimmunity [15], while a separate study showed no difference in 25OHD levels of newly diagnosed children with type 1 diabetes compared to those of their healthy siblings [16]. Another prospective study [17] demonstrated an association of low 25OHD with presence of pancreatic islet autoantibodies but not with progression to type 1 diabetes. It is important to note that prospective studies of the effect of 25OHD levels on type 1 diabetes starting before the occurrence of islet autoantibodies are less susceptible to reverse causation compared to other study types in the field.

The link between vitamin D and type 1 diabetes may be mediated by the effects of vitamin D on the immune system. There is evidence that the active form of vitamin D, 1,25-dihydroxyvitamin D, is an immune modulator, reducing activation of the immune system [18]. It has been argued that vitamin D has also nonimmunologic effects on the pancreas and thus may directly influence beta-cell function [19].

The reason the epidemiological associations have not led to changes in clinical care is because the aforementioned observational studies can be hampered by confounding or reverse causation. In this scenario, confounding could cause a spurious association if vitamin D and type 1 diabetes are actually linked through an unobserved association with another disease determinant, such as latitude or ethnicity. According to the "sunshine hypothesis" [20], the increasing prevalence of this disease can be explained by the fact that less time is spent outdoors and there is less exposure to ultraviolet radiation, leading to vitamin D deficiency. This

hypothesis basically comes from the observation that countries closer to the equator have lower rates of type 1 diabetes [21] and from the seasonal variations in its incidence, with most cases being diagnosed in winter [22]. Also, patients affected by type 1 diabetes are more sensitive to ultraviolet radiation and have less skin pigmentation than healthy controls [23], which could suggest that reduced sun exposure might account for the low level of vitamin D in these patients. Reverse causation can represent another limitation of the observational studies. For instance, the onset of type 1 diabetes may lead individuals to remain indoors because of fear of hypoglycemia with physical activity, which would decrease vitamin D levels due to reduced sunlight exposure. Given these findings and in the absence of evidence from randomized controlled trials (RCTs), strongly implicating vitamin D as a causal factor in etiology of type 1 diabetes in humans remains difficult.

However, support for a causal role for vitamin D in type 1 diabetes would have important public health implications. The prevalence of this disease under the age of 14 years old is expected to rise by 3% annually worldwide [24]. In addition, the prevalence of vitamin D insufficiency is estimated to be 43% in the general population [25] and is also increasing [26]. There is a significant economic burden and long-term care costs associated with type 1 diabetes [27]. In contrast, an annual supply of 1,000 IU vitamin D supplements satisfying the Institute of Medicine's intake guidelines for sufficiency [28] costs approximately $30 to $40. Therefore, ensuring vitamin D sufficiency among individuals at high risk for type 1 diabetes may be explored as a cost-effective approach to reduce risk, if clinical trial evidence supports a role for vitamin D administration in the prevention of this disease. Despite this, the considerable controversy in the available epidemiological data has prevented the conduct of large-scale RCTs, also because such trials would be expensive and reliant upon funding of the public purse, since vitamin D cannot be patented. Therefore, it is necessary to better understand the causal relationship between vitamin D and type 1 diabetes in humans, in order to provide evidence to support or not a costly RCT.

Mendelian randomization (MR) is a method in genetic epidemiology which uses genetic variants reliably associated with exposures of interest to estimate causal associations between a given biomarker, such as vitamin D, and disease. These studies have orthogonal assumptions to conventional techniques and are arguably less prone to reverse causation since disease states usually do not change the germline DNA sequences [29]. Most importantly, MR can limit confounding, since genotypes are randomly assorted at meiosis [30]. In this regard, the MR framework can be compared to that of an RCT because the random assortment of genetic variants replicates the random allocation of study participants to different therapeutic arms. Lastly, since genetic variants remain stable over a lifetime, MR studies provide insights from a lifetime of genetically altered biomarker levels, which, in this case, is lowered 25OHD levels. Therefore, we elected to perform an MR study using single nucleotide polymorphisms (SNPs) from the largest genome-wide association study (GWAS) for vitamin D to date on 443,734 Europeans [31], and their effects on type 1 diabetes from a recent large meta-analysis of GWAS on 9,358 cases and 15,705 controls [32].

## Methods

The present study did not follow a prespecified analysis plan or protocol. Ethics approval was not required for this study. We provide a completed STrengthening the Reporting of OBservational studies in Epidemiology (STROBE) checklist for the study (see S1 STROBE Checklist).

### Genetic variants associated with vitamin D

To assess whether genetically lowered vitamin D levels are associated with increased odds of type 1 diabetes, we identified, among the conditionally independent SNPs of a large 25OHD

GWAS meta-analysis ($n$ = 443,734) [31], the lead common independent SNPs associated with 25OHD. To satisfy the first MR assumption, which requires that the instrument (SNP) robustly associates with the exposure (25OHD level), we chose as instruments SNPs which were associated with 25OHD levels at a level of genome-wide significance ($p < 6.6 \times 10^{-9}$). We selected only common SNPs among the conditionally independent SNPs, to ensure that our instruments are truly in linkage equilibrium, since the $r^2$ as metric of linkage disequilibrium (LD) is less accurate for rare variants [33]. Also, rare variants with minor allele frequency (MAF) <0.5% were generally absent in the type 1 diabetes GWAS. We extracted estimates of the effects of these 25OHD-associated variants on type 1 diabetes from a large GWAS meta-analysis of 12 European studies, totaling 9,358 cases and 15,705 controls [32]. Details on the demographics of the cohorts participating in the 25OHD and type 1 diabetes GWAS can be found in the respective publications [31,32]. We also estimated the total variance explained in 25OHD by our genetic instruments. The variance explained for a given SNP was calculated using the formula: variance explained = $2\beta^2 f (1 -f)$, where $\beta$ and $f$ denote the effect of the SNP on 25OHD level and the MAF, respectively.

For 25OHD-related variants not directly present in the type 1 diabetes GWAS, we selected proxy SNPs in LD ($r2 > 0.7$) using the LDproxy function in LD link (https://ldlink.nci.nih.gov) and all EUR populations from 1,000 genomes phase 3.

## Mendelian randomization analyses

We performed a main analysis using an inverse-variance weighted (IVW) two-sample MR to estimate the effect of a 1 standard deviation (SD) increase in standardized natural log-transformed 25OHD on type 1 diabetes susceptibility, using previously described methods [34]. Specifically, the effect of each variant on risk of type 1 diabetes was weighted by its effect on 25OHD using the Wald ratio method. These individual MR estimates were then combined in a random effect inverse-variance meta-analysis. To help the interpretation of our MR results, we also expressed our MR estimates as the effect of a given nmol/l change in 25OHD levels (corresponding to 1 SD in standardized natural log-transformed 25OHD) in vitamin D–deficient, vitamin D–insufficient, and vitamin D–sufficient individuals, defined as individuals having 25OHD levels at the clinically relevant thresholds of 25, 50, and 70 nmol/l, respectively.

The second MR assumption requires that the instruments (SNPs) are not associated with phenotypes that could confound the association between the exposure and the outcome. As a sensitivity analysis, MR estimates excluding variants associated with potential confounders were calculated. To do so, we queried in the PhenoScanner database each 25OHD-related SNP used as instrument, in order to identify genetic variants associated with GWAS traits that are potential confounders or could introduce horizontal pleiotropy in the exposure-outcome association (associating variants). We considered positive associations whenever the GWAS $p$-value of the variant for a trait was below the nominal $p$-value Bonferroni-corrected for the number of genetic variants ($p < 0.05/69 = 0.001$). Associations with specific traits appearing more than once per SNP in our Phenoscanner search (meaning that they were present in more than one GWAS) were counted as single entries. Certain traits clustered into larger trait categories; therefore, we grouped these into 14 trait categories (S2 Table). We then counted the number of SNP-trait associations and looked for enrichments of each of the 25OHD SNPs in traits clustering into the 14 categories. While we did not detect associations of our MR SNPs with traits considered as confounders in the association between vitamin D and type 1 diabetes (and therefore eliminated the risk of horizontal pleiotropy), we selected the top 3 trait categories (S1 and S2 Tables), defined as the categories that accumulated the most SNP-trait associations, and performed sensitivity analyses excluding SNPs mapping in each of these 3 categories

respectively. We additionally performed a sensitivity analysis excluding SNPs associated with type 2 diabetes and related traits. The rationale for performing sensitivity analyses by omitting SNPs associated with these traits is that the latter might explain how the SNPs act upon 25OHD levels or explain downstream effects (vertical pleiotropy).

The third MR assumption is that the genetic variants must not be associated with the outcome through pathways other than the exposure of interest (referred to as exclusion restriction assumption) [34]. In the context of MR, horizontal pleiotropy refers to a scenario in which this assumption is breached. To test this assumption, we applied different approaches which account for potential pleiotropic effects. First, we tested for heterogeneity of the SNPs used as instruments and generated MR estimates omitting SNPs appearing as outliers [35]. We then applied MR-Egger regression to account for potentially unmeasured pleiotropy [36]. This method consists of a weighted linear regression of the SNP-type 1 diabetes susceptibility on the SNP-25OHD associations. This allows the estimation of an intercept as a measure of the average pleiotropic effect and produces a slope coefficient as a robust to pleiotropy MR estimate. MR-Egger allows a weakening of the exclusion restriction assumption and requires the association of each variant with 25OHD not be correlated with its pleiotropic effect (known as the InSIDE assumption). Additionally, we performed a weighted median analysis [37], which weights individual MR estimates by their precision. This approach relies on the fact that estimates from SNPs without pleiotropic effects are more likely to converge toward the median, while we could expect that pleiotropy will introduce heterogeneity and result in relative outliers. This method provides reliable results when less than 50% of the total weight is coming from variants with pleiotropic effects. Finally, we used a similar approach to the previous, but that relies on a mode-based estimate rather than the median, allowing for even the majority of SNPs to be pleiotropic [38]. Lastly, to further ensure that our estimates were not influenced by pleiotropy, we repeated the MR analysis using only 6 genetic variants for 25OHD levels identified in a previous GWAS and also undertook a separate analysis using 4 among these 6 SNPs, which lie in or next to the *DHCR7*, *CYP2R1*, *GC*, *CYP24A1* genes which directly regulate vitamin D synthesis or degradation [39]. By undertaking this variety of sensitivity analyses, and comparing results using these approaches, each one with different underlying assumptions, we ensured that it is unlikely that our findings are biased by of pleiotropy.

We therefore used the MendelianRandomization R package [40], and its default parameters, to compute the 4 different MR estimates [IVW, weighted median, random-effects MR-Egger, and mode-based estimate (MBE)] for the main analysis, including all SNPs, and in the sensitivity analyses omitting SNPs associated with confounders, and proxy SNPs. Notably, the "random" model was selected in the MR Egger and IVW methods, given the presence of heterogeneity in our instruments. The "penalized" parameter also penalized variants with heterogeneous causal estimates. We used the same package to generate scatter plots to compare the MR estimates using different methods. As an additional control for pleiotropy, we applied the global test, outlier test, and distortion test using the MR pleiotropy residual sum and outlier (MR-PRESSO) R package [35]. Specifically, the global test detects horizontal pleiotropy among the MR instruments; the outlier test corrects for horizontal pleiotropy via outlier removal; the distortion test identifies significant distortion in the causal estimates before and after outlier removal.

The type 1 diabetes GWAS was restricted to individuals of European descent, similar to the 25OHD GWAS, in order to limit bias from population stratification. Finally, we undertook power calculations using the method published by Brion and colleagues [41] to test whether our study was adequately powered to detect clinically relevant changes in type 1 diabetes risk. To do this, we set the alpha level to 0.05 and used the estimate of the variance explained by the 25OHD-related SNPs produced by the aforementioned formula.

## Results

### SNP selection and genetic effect sizes on 25OHD

We used 69 lead common independent SNPs (**S1 Table**), explaining 3.1% of the variance in 25OHD levels, as instruments in our MR studies. Eight out of the 69 SNPs were absent in the type 1 diabetes GWAS and were replaced by proxies in high LD ($r^2 > 0.7$) (**Table 1**). Our PhenoScanner search did not reveal any associations, at a Bonferroni-corrected threshold, with any known potentially pleiotropic pathways influencing these 2 outcomes. However, we observed an enrichment in SNPs associated with certain trait categories, in particular blood cell counts, body composition, and serum lipid traits (**S1 Table**). Specifically, among 1,641 SNP-trait associations with 494 individual traits found by our PhenoScanner search, we observed 246 associations with blood cell counts, 250 associations with body composition traits, and 213 associations with serum lipids. A full description of the SNP-trait associations is provided in **S2 Table.** We then undertook sensitivity analyses excluding SNPs clustering in each of these 3 trait categories and also excluding SNPs associated with type 2 diabetes–related traits.

The results of the MR analyses are shown in **Figs 1–3** and **Table 1.** We did not find evidence supporting a causal association between 25OHD levels and risk of type 1 diabetes (IVW MR OR = 1.09, 95% CI: 0.86 to 1.40, $p$ = 0.48 per 1 SD decrease in standardized log-transformed 25OHD). Similar results were obtained using the other 3 MR methods. We estimated that a 1 SD change in standardized natural-log transformed 25OHD levels corresponds to a change in 25OHD levels of 40.9 nmol/l in vitamin D–sufficient individuals (defined as individuals having 25OHD levels of 70 nmol/l), of 29.2 nmol/l in vitamin D–insufficient individuals (defined as having 25OHD levels of 50 nmol/l), and of 14.6 nmol/l in vitamin D–deficient individuals (defined as having 25OHD levels of 25 nmol/l). Of note, a 29.2-nmol/l change in 25OHD levels is comparable to the 21.2 nmol/L mean increase in 25OHD levels conferred by taking daily 400 IU of cholecalciferol, the amount of vitamin D most often found in vitamin D supplements [42].

There was evidence of heterogeneity across the individual MR estimates derived from the 69 SNPs in the main MR analyses (IVW Q > 100, $p$-value ≤ 0.002 in both MR studies). The intercept estimated from the MR-Egger regression was centered around zero (−0.009, $p$-value = 0.1) and did not provide strong evidence for unbalanced horizontal pleiotropy. However, using MR-PRESSO, we found evidence for pleiotropy ($p$-value global test 0.001). The MR estimates for type 1 diabetes did not alter the inference of the results after removing 1 outlier SNPs with increased evidence of pleiotropic effects (**S3 Table**). Moreover, we conducted sensitivity analyses after removing subsequently 8 proxy SNPs, 35 SNPs associated with blood cell counts, 23 SNPs associated with body composition, and 22 SNPs associated with lipid phenotypes for both outcomes, and also 9 type 2 diabetes SNPs (**S1 and S2 Tables**). All sensitivity analyses yielded results similar with those of the main analyses. The results of these sensitivity analyses appear in **Figs 1–3.** Finally, using only the 6 SNPs for 25OHD identified in a previous GWAS [39], we obtained similar estimates (IVW MR OR = 0.96, 95% CI: 0.58 to 1.58, $p$ = 0.86 per 1 SD decrease in standardized log-transformed 25OHD) (**S4 Table**). When we limited our MR instruments to only 4 among these 6 SNPs (which involve genes with direct role in vitamin D synthesis and metabolism explaining 2.4% of the variance in 25OHD levels), the results were similar (IVW MR OR = 0.92, 95% CI:0.53 to 1.62, $p$ = 0.78).

Based on a sample size of 25,063 individuals and setting alpha to 0.05, and the variance explained to 3.1%, our study had a power of 80% power to exclude effects on type 1 diabetes as small as an OR of 1.23 per 1 SD change in 25OHD on the log scale and a power of 100% to exclude effects as small as an OR of 1.4.

**Table 1. Results of the MR study testing causal association between low 25OHD and type 1 diabetes.**

| Full analysis with all SNPs (N = 69) | | | | | | | |
|---|---|---|---|---|---|---|---|
| Analysis | Odds ratio | CI lower | CI upper | p-value | Egger intercept | Intercept p-value | |
| Inverse variance weighted (random) | 1.093 | 0.855 | 1.396 | 0.478 | | | heterogeneity: Q = 113.9; p < 0.001 |
| Weighted median | 1.139 | 0.882 | 1.472 | 0.318 | | | |
| MR-Egger (random) | 1.314 | 0.948 | 1.821 | 0.101 | −0.009435201 | 0.099 | heterogeneity Q = 109.5; p < 0.001; $I^2$ univariable MR-Egger = 0.9931612 |
| MBE (simple,1) | 1.143 | 0.902 | 1.447 | 0.268 | | | |
| **excluding proxy SNPs (N = 61)** | | | | | | | |
| Analysis | Odds ratio | CI lower | CI upper | p-value | Egger intercept | Intercept p-value | |
| Inverse variance weighted (random) | 1.090 | 0.844 | 1.407 | 0.510 | | | heterogeneity: Q = 101.6; p < 0.001 |
| Weighted median | 1.139 | 0.875 | 1.483 | 0.334 | | | |
| MR-Egger (random) | 1.261 | 0.902 | 1.764 | 0.175 | −0.007952573 | 0.191 | heterogeneity Q = 98.8 p < 0.001; $I^2$ univariable MR-Egger = 0.9938547 |
| MBE (simple,1) | 1.121 | 0.888 | 1.415 | 0.337 | | | |
| **excluding lipids SNPs (N = 47)** | | | | | | | |
| Analysis | Odds ratio | CI lower | CI upper | p-value | Egger intercept | Intercept p-value | |
| Inverse variance weighted (random) | 1.050 | 0.818 | 1.347 | 0.703 | | | heterogeneity: Q = 70.0; p = 0.00127 |
| Weighted median | 1.132 | 0.881 | 1.456 | 0.332 | | | |
| MR-Egger (random) | 1.260 | 0.918 | 1.730 | 0.153 | −0.01102054 | 0.077 | heterogeneity Q = 65.5; p = 0.0246; $I^2$ univariable MR-Egger = 0.9951814 |
| MBE (simple,1) | 1.109 | 0.880 | 1.398 | 0.381 | | | |
| **excluding blood cell SNPs (N = 34)** | | | | | | | |
| Analysis | Odds ratio | CI lower | CI upper | p-value | Egger intercept | Intercept p-value | |
| Inverse variance weighted (random) | 1.171 | 0.741 | 1.852 | 0.498 | | | heterogeneity: Q = 72.2; p < 0.001 |
| Weighted median | 1.577 | 0.948 | 2.623 | 0.079 | | | |
| MR-Egger (random) | 1.738 | 0.861 | 3.508 | 0.123 | −0.01539115 | 0.151 | heterogeneity Q = 67.8; p < 0.001; $I^2$ univariable MR-Egger = 0.9878622 |
| MBE (simple,1) | 1.212 | 0.785 | 1.871 | 0.386 | | | |
| **excluding body composition SNPs (N = 46)** | | | | | | | |
| Analysis | Odds ratio | CI lower | CI upper | p-value | Egger intercept | Intercept p-value | |
| Inverse variance weighted (random) | 1.111 | 0.882 | 1.400 | 0.372 | | | heterogeneity: Q = 56.7; p = 0.1124 |
| Weighted median | 1.161 | 0.885 | 1.522 | 0.281 | | | |
| MR-Egger (random) | 1.418 | 1.066 | 1.884 | 0.016 | −0.01466228 | 0.009 | heterogeneity Q = 49.2; p = 0.2737; $I^2$ univariable MR-Egger = 0.9950168 |
| MBE (simple,1) | 1.218 | 0.950 | 1.562 | 0.120 | | | |
| **excluding diabetes SNPs (N = 60)** | | | | | | | |
| Analysis | Odds ratio | CI lower | CI upper | p-value | Egger intercept | Intercept p-value | |
| Inverse variance weighted (random) | 1.112 | 0.862 | 1.436 | 0.414 | | | heterogeneity: Q = 101.7; p < 0.001 |
| Weighted median | 1.149 | 0.882 | 1.497 | 0.304 | | | |
| MR-Egger (random) | 1.354 | 0.971 | 1.888 | 0.074 | −0.01072743 | 0.077 | heterogeneity Q = 96.5; p = 0.0011; $I^2$ univariable MR-Egger = 0.9939633 |
| MBE (simple,1) | 1.179 | 0.935 | 1.487 | 0.165 | | | |

The OR are per 1 standard deviation decrease in standardized log-transformed 25OHD.

25OHD, 25-hydroxyvitamin D; CI, confidence interval; MBE, mode-based estimate; MR, Mendelian randomization; SNP, single nucleotide polymorphism.

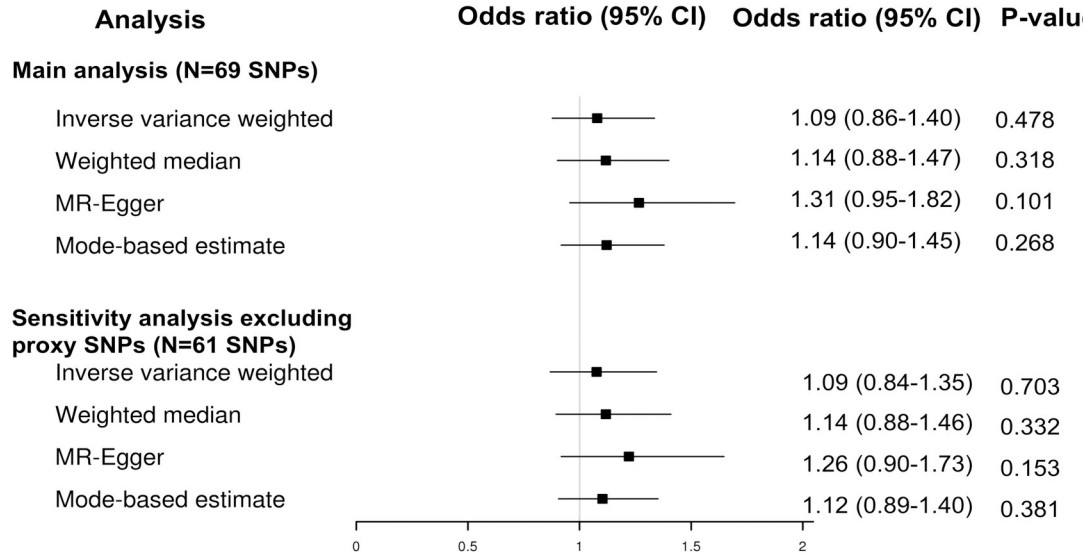

| Analysis | Odds ratio (95% CI) | Odds ratio (95% CI) | P-value |
|---|---|---|---|
| **Main analysis (N=69 SNPs)** | | | |
| Inverse variance weighted | | 1.09 (0.86-1.40) | 0.478 |
| Weighted median | | 1.14 (0.88-1.47) | 0.318 |
| MR-Egger | | 1.31 (0.95-1.82) | 0.101 |
| Mode-based estimate | | 1.14 (0.90-1.45) | 0.268 |
| **Sensitivity analysis excluding proxy SNPs (N=61 SNPs)** | | | |
| Inverse variance weighted | | 1.09 (0.84-1.35) | 0.703 |
| Weighted median | | 1.14 (0.88-1.46) | 0.332 |
| MR-Egger | | 1.26 (0.90-1.73) | 0.153 |
| Mode-based estimate | | 1.12 (0.89-1.40) | 0.381 |

**OR for type 1 diabetes per 1 SD decrease in 25OHD on the log scale**

**Fig 1. Forest plot of the MR study investigating the effect of 25OHD on type 1 diabetes.** Forest plot of the main study and of the sensitivity analysis excluding proxy SNPs. 25OHD, 25-hydroxyvitamin D; MR, Mendelian randomization; SNP, single nucleotide polymorphism.

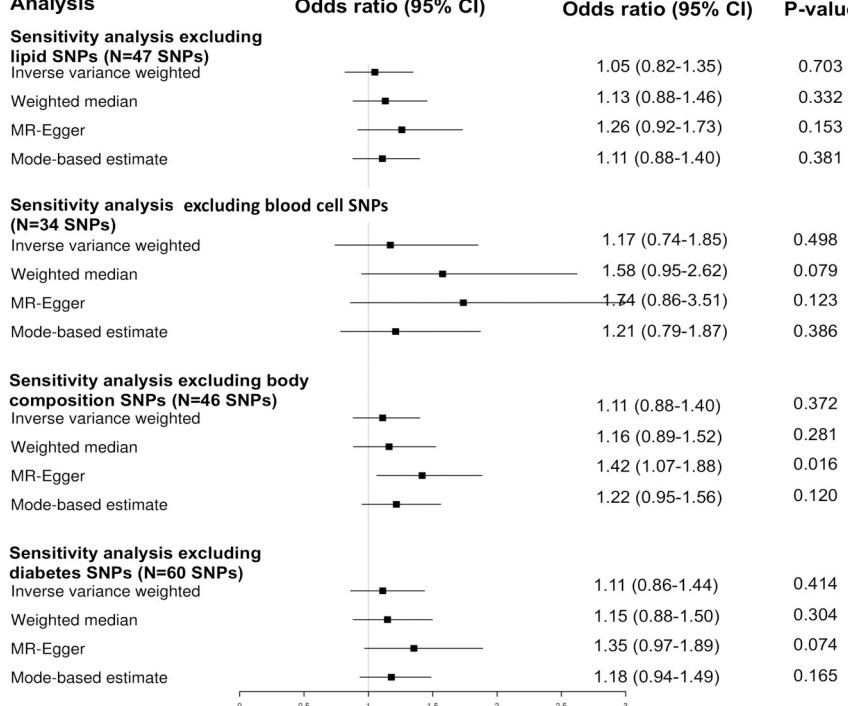

| Analysis | Odds ratio (95% CI) | Odds ratio (95% CI) | P-value |
|---|---|---|---|
| **Sensitivity analysis excluding lipid SNPs (N=47 SNPs)** | | | |
| Inverse variance weighted | | 1.05 (0.82-1.35) | 0.703 |
| Weighted median | | 1.13 (0.88-1.46) | 0.332 |
| MR-Egger | | 1.26 (0.92-1.73) | 0.153 |
| Mode-based estimate | | 1.11 (0.88-1.40) | 0.381 |
| **Sensitivity analysis excluding blood cell SNPs (N=34 SNPs)** | | | |
| Inverse variance weighted | | 1.17 (0.74-1.85) | 0.498 |
| Weighted median | | 1.58 (0.95-2.62) | 0.079 |
| MR-Egger | | 1.74 (0.86-3.51) | 0.123 |
| Mode-based estimate | | 1.21 (0.79-1.87) | 0.386 |
| **Sensitivity analysis excluding body composition SNPs (N=46 SNPs)** | | | |
| Inverse variance weighted | | 1.11 (0.88-1.40) | 0.372 |
| Weighted median | | 1.16 (0.89-1.52) | 0.281 |
| MR-Egger | | 1.42 (1.07-1.88) | 0.016 |
| Mode-based estimate | | 1.22 (0.95-1.56) | 0.120 |
| **Sensitivity analysis excluding diabetes SNPs (N=60 SNPs)** | | | |
| Inverse variance weighted | | 1.11 (0.86-1.44) | 0.414 |
| Weighted median | | 1.15 (0.88-1.50) | 0.304 |
| MR-Egger | | 1.35 (0.97-1.89) | 0.074 |
| Mode-based estimate | | 1.18 (0.94-1.49) | 0.165 |

**OR for type 1 diabetes per 1 SD decrease in 25OHD on the log scale**

**Fig 2. Forest plot of the MR sensitivity analyses excluding SNPs with possible pleiotropic effects.** The odds ratios for type 1 diabetes are reported for a 1 standard deviation decrease in 25OHD on the log scale. 25OHD, 25-hydroxyvitamin D; CI, confidence intervals; MR, Mendelian randomization; OR, odds ratio; SNP, single nucleotide polymorphism.

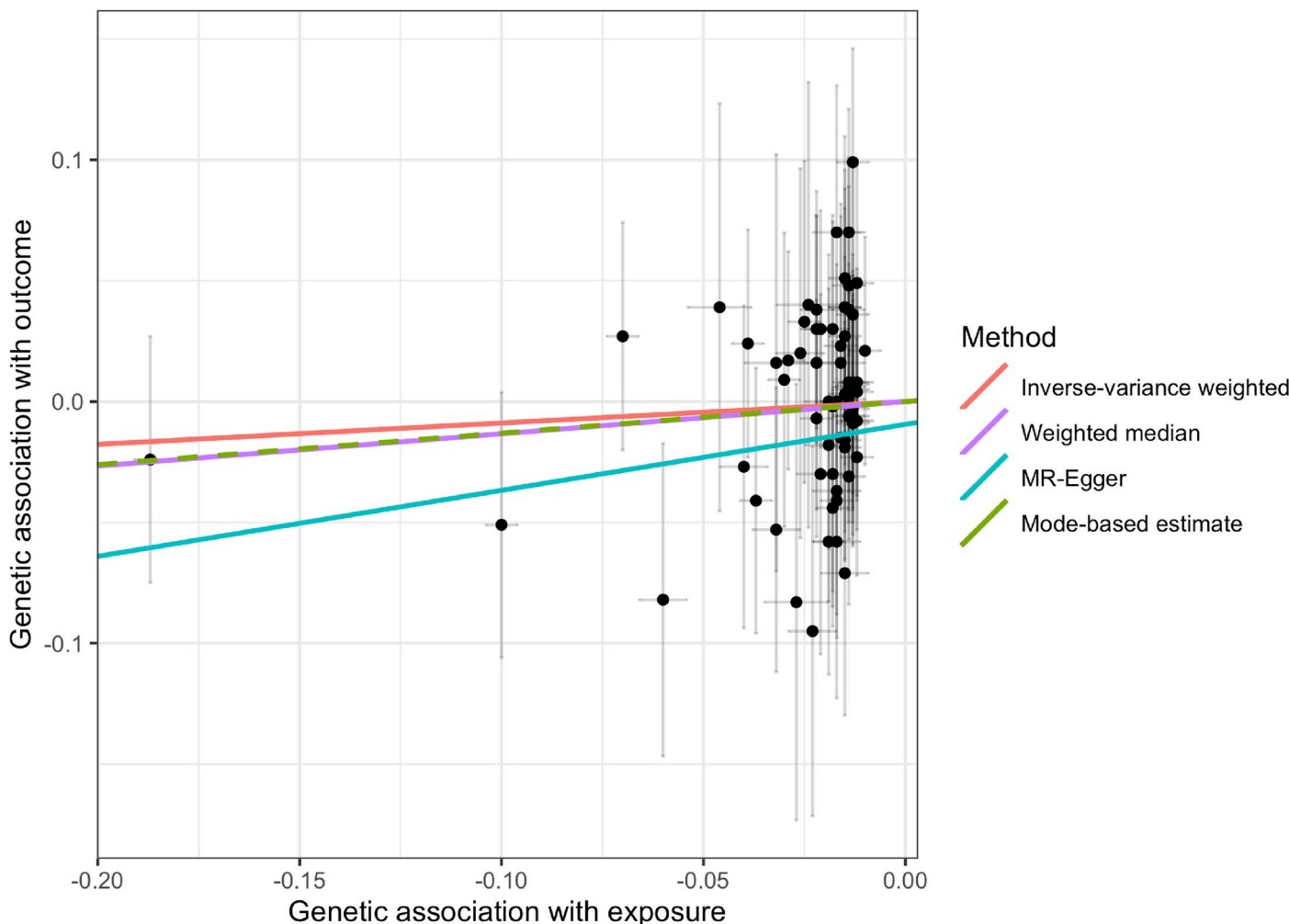

**Fig 3. Scatter plot of the main MR study investigating the effect of 25OHD on type 1 diabetes.** The x-axis represents the genetic association with 25OHD levels; the y-axis represents the genetic association with risk of type 1 diabetes. Each line represents a different MR method. 25OHD, 25-hydroxyvitamin D; MR, Mendelian randomization.

## Discussion

The results of this MR study do not support a causal association between a genetically determined change in 25OHD levels—comparable to the effect of a conventional dose of vitamin D supplementation in a vitamin D–insufficient individual—and risk of type 1 diabetes. Our findings confidently exclude large effects (OR >1.40), given the sufficient statistical power and the consistency of the estimates using different MR methods and in the sensitivity analyses. However, the relatively large confidence intervals of our MR result suggests that small effects cannot be excluded, and further evidence is needed to investigate such effects.

Our results are generally in contrast with observational evidence, largely from northern European populations, supporting a role of low 25OHD in type 1 diabetes [12,13]. They are consistent with those of 3 prospective studies [14,15,17], but, due to the nature of the MR analysis, are less prone to confounding and reverse causation. Our findings are also in agreement with those of a recent observational study in individuals residing in a solar-rich environment [9].

No previous MR studies have been performed to investigate this research hypothesis. Although associations between genetic variants influencing 25OHD levels and risk of type 1 diabetes have been studied in previous works by Cooper and colleagues [43] and Thorsen and colleagues [44], the authors of these papers reported straightforward associations of these variants with disease status in cohorts of type 1 diabetes cases and controls. More recently, a phenome-wide association study by Meng and colleagues [45] examined associations between a polygenic risk score for 25OHD levels comprising 6 SNPs and 920 phenotypes in UK Biobank, among which was type 1 diabetes. The authors found no evidence for positive association of 25OHD with type 1 diabetes. The 3 aforementioned studies do not fulfill the criteria of an MR approach. Also, since these studies were published, our knowledge on vitamin D genetics was substantially expanded with the publication of large vitamin D GWAS meta-analyses [31], providing more precise association tests and a new set of 25OHD SNPs explaining a larger portion of the variance in 25OHD levels.

Our MR approach has several strengths. First, its design decreases potential confounding or reverse causation which are present in observational studies. Eliminating reverse causation is important since type 1 diabetes may be characterized by a preclinical phrase, which renders it difficult to determine whether an exposure precedes the pathological changes to the pancreatic beta cell. Our analysis also captures lifetime risk of type 1 diabetes due to genetically decreased vitamin D, which again is important since a single vitamin D measurement is unlikely to be an accurate predictor of a disease that manifests later in life. Lastly, by employing the two-sample MR approach, we were able to test the effect of vitamin D in a large cohort of type 1 diabetes patients ($N$ = 9,358 type 1 diabetes cases and 15,705 controls). Such two-sample approaches have statistical power comparable to an approach using individual-level data [46]; however, few cohorts have accrued as many cases and controls for type 1 diabetes.

Our analysis also has limitations worth consideration. While we undertook multiple steps to examine pleiotropy, residual bias is possible since the exact function of most of these SNPs is unknown. Even in the case that pleiotropy was not properly accounted for by MR egger (if the INSIDE assumption was violated by some of the SNPs used as instruments), the fact that we obtained consistent results using another 2 pleiotropy robust MR methods and in sensitivity analyses excluding SNPs with pleiotropic effects is reassuring. Canalization is another mechanism which may bias MR results toward the null. In this scenario, the effects of genetically reduced 25OHD levels on pathophysiology of type 1 diabetes may have been mitigated by physiologic compensation [47], which, in this scenario, could lead to an increase conversion of 25OHD to its active form 1,25 dihydroxyvitamin D. In this regard, it might also be possible that the immune effects of vitamin D on the beta cell are correlated to the levels of the active form of vitamin D (1,25 dihydroxyvitamin D), which are weakly correlated with 25OHD levels. Thus, although genetically lowered total 25OHD levels do not appear to be associated with increased risk of type 1 diabetes, our study still leaves open the possibility that reduced lifelong 1,25 dihydroxyvitamin D is indeed associated with type 1 diabetes. However, changes in 25OHD levels are commonly measured to diagnose vitamin D insufficiency and monitor response to vitamin D supplements, in contrast to 1,25 dihydroxyvitamin D levels, which are unstable, have a short half-life, and are not routinely measured in clinical practice.

MR studies can assess the relationship between a biomarker and a disease only at the time point in the life course where the genetic variant has been associated with the biomarker. This may be important if the genetic determinants of 25OHD levels are different in adulthood when compared to childhood, despite lack of evidence from the literature supporting this for 25OHD levels. In this regard, our study could not exclude effects of intrauterine exposure to lower 25OHD levels in the risk of type 1 diabetes in the offspring which have been examined in observational studies [48,49]. Despite the fact that the SNPs used as instruments in our MR

were extracted from GWAS in Europeans, the populations of both GWAS were not homogenous in terms of geographic location. It has been shown that there could be some gene-environment interaction in the effect of SNPs in the vitamin D receptor gene on type 1 diabetes risk [50]. Specifically, the effect of these SNPs on type 1 diabetes varies with levels of ultraviolet irradiation, suggesting that the impact in vitamin D–deficient groups differs from that in non-deficient population. This raises a possibility of gene-environment interaction for SNPs affecting 25OHD levels and of nonlinear effects of these SNPs on risk of type 1 diabetes, but two-sample MR studies can only assess linear associations. As such, they do not assess either the effects on disease of having levels of a biomarker in the extremes of the normal distribution. However, it is difficult to assess the impact of the above limitations in our MR study, given the fact that prospective studies have reported a linear relationship between doses of vitamin D supplementation and risk of type 1 diabetes [6]. Our MR results cannot be generalized to non-Europeans and potentially to Europeans residing in different geographic areas than those of the participants in the vitamin D and type 1 diabetes GWAS; and although we assured that both exposure and outcome GWAS were restricted to participants of European ancestry, residual confounding from population stratification cannot be completely excluded in a two-sample MR setting, due to ethnic differences in the exposure and outcome GWAS populations [51]. Finally, there is a possibility that our MR power calculation was overestimated due to overfitting caused by the fact that the variance explained by the vitamin D SNPs was calculated in the same population as the vitamin D GWAS. Unfortunately, we do not have access to an independent cohort where the amount of variance explained by those SNPs could be estimated.

In conclusion, our results identified no large impact of a genetically determined reduction in 25OHD levels on type 1 diabetes risk. This provides critical insight into a complex disease that remains poorly understood. Our findings imply that the observational associations between 25OHD and risk of type 1 diabetes might be due to environmental confounders, such as latitude, which is correlated with exposure to sun and skin pigmentation, but this needs to be investigated in further studies. Since small effects cannot be excluded, our findings need to be confirmed by large RCTs testing the effects of vitamin D supplements in type 1 diabetes, or future MR evidence based on larger GWAS samples for 25OHD and type 1 diabetes risk.

## Supporting information

**S1 STROBE checklist. A completed STrengthening the Reporting of OBservational studies in Epidemiology (STROBE) checklist for the study.**
(DOCX)

**S1 Table. List of the 69 conditionally independent common variants used as instruments in the Mendelian randomization studies.**
(XLSX)

**S2 Table. PhenoScanner traits associated with the 69 SNPs used as instruments in the MR studies.**
(XLSX)

**S3 Table. Horizontal pleiotropy assessment using MR-PRESSO for the vitamin D-type 1 diabetes MR.**
(XLSX)

**S4 Table. MR sensitivity analysis with the 6 common variants from the Jiang et al. GWAS.**
(XLSX)

## Author Contributions

**Conceptualization:** Despoina Manousaki, Constantin Polychronakos, J Brent Richards.

**Data curation:** Despoina Manousaki, Ruth E. Mitchell, Stephanie Ross.

**Formal analysis:** Despoina Manousaki.

**Funding acquisition:** Despoina Manousaki, J Brent Richards.

**Methodology:** Despoina Manousaki, Adil Harroud, Ruth E. Mitchell, Vince Forgetta, George Davey Smith.

**Project administration:** Despoina Manousaki, J Brent Richards.

**Resources:** Vince Forgetta, Nicholas J. Timpson, J Brent Richards.

**Software:** Despoina Manousaki, Stephanie Ross, Vince Forgetta, George Davey Smith.

**Supervision:** Vince Forgetta, Nicholas J. Timpson, Constantin Polychronakos, J Brent Richards.

**Validation:** Despoina Manousaki, Adil Harroud, George Davey Smith, J Brent Richards.

**Visualization:** Despoina Manousaki, Adil Harroud, Nicholas J. Timpson.

**Writing – original draft:** Despoina Manousaki.

**Writing – review & editing:** Despoina Manousaki, Adil Harroud, Ruth E. Mitchell, Stephanie Ross, Vince Forgetta, Nicholas J. Timpson, George Davey Smith, Constantin Polychronakos, J Brent Richards.

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
