## [Editor Report · Decision Letter 0]

26 Aug 2020

Dear Dr Manousaki, 

Thank you for submitting your manuscript entitled "Genetically decreased vitamin D and risk of type 1 diabetes: A Mendelian randomization study" for consideration by PLOS Medicine.

Your manuscript has now been evaluated by the PLOS Medicine editorial staff and I am writing to let you know that we would like to send your submission out for external assessment.

Kind regards,

Richard Turner, PhD

Senior editor, PLOS Medicine

rturner@plos.org

---

## [Decision Letter · Decision Letter 1]

21 Sep 2020

Dear Dr. Manousaki,

Thank you very much for submitting your manuscript "Genetically decreased vitamin D and risk of type 1 diabetes: A Mendelian randomization study" (PMEDICINE-D-20-04125R1) for consideration at PLOS Medicine. 

Your paper was discussed among the editors and sent to independent reviewers, including a statistical reviewer. The reviews are appended at the bottom of this email and any accompanying reviewer attachments can be seen via the link below:

[LINK]

In light of these reviews, we will not be able to accept the manuscript for publication in the journal in its current form, but we would like to invite you to submit a revised version that addresses the reviewers' and editors' comments fully. You will appreciate that we cannot make a decision about publication until we have seen the revised manuscript and your response, and we plan to seek re-review by one or more of the reviewers. 

We hope to receive your revised manuscript by Oct 12 2020 11:59PM. Please email us (plosmedicine@plos.org) if you have any questions or concerns.

Please let me know if you have any questions. Otherwise, we look forward to receiving your revised manuscript in due course. 

Sincerely,

Richard Turner, PhD

rturner@plos.org

We note that one reviewer mentions STROBE-MR. As far as we are aware, this has not yet been published in peer-reviewed form, and so we are not requesting that authors of relevant studies use that guideline. 

We suggest adding "levels" to the title.

Please add a new final sentence to the "Methods and findings" subsection of your abstract, quoting 2-3 of the study's main limitations. 

Please add "Our findings suggest that ..." or similar to begin the "Conclusions" subsection of your abstract. 

After the abstract, please add a new and accessible "author summary" section in non-identical prose. You may find it helpful to consult one or two recent research papers published in PLOS Medicine to get a sense of the preferred style. 

Throughout the paper, please adapt reference call-outs to the following style: "... [8]." (i.e., consistently preceded by a space and preceding punctuation, and without spaces within the square brackets).

Early in the methods section of your main text, please state whether the study had a prespecified analysis plan or protocol, and if so attach the document(s) as a supplementary file(s), referred to in the text. Please highlight analyses that were not prespecified. 

Please note in the Methods section that ethics approval was not required for this study (or quote the authority and ethics approval). 

At the start of the Discussion section, please rephrase "... failed to support ...". 

Throughout the text, please quote p values alongside 95% CI, where available. 

Please remove the information on funding from the end of the main text. In the event of publication, this information will appear in the article metadata, via entries in the submission form. 

Noting reference 1 and some others, e.g., reference 26 and 27, please ensure that all references quote full access details. 

Please spell out the institutional author name for reference 22, and add access details. 

Please add a completed checklist for the most appropriate reporting guideline, which we suspect will be STROBE, as a supplementary document, referred to in the methods section (e.g., "See S1_STROBE_Checklist"). In the checklist, please refer to individual items by section (e.g., "Methods") and paragraph number rather than by line or page numbers, as the latter generally change in the event of publication. 

Comments from the reviewers:

*** Reviewer #1: 

This paper looks at the association between Vitamin D and Type 1 diabetes, through a Mendelian randomization framework. It is not a completely novel result, but it is confirmation of the null effect previously published in "Phoneme-wide Mendelian-randomization study of genetically determined vitamin D on multiple health outcomes using the UK Biobank study" https://academic.oup.com/ije/article/48/5/1425/5569493. Unlike that paper, this one focuses only on a single outcome, and so does a range of sensible robustness checks on the result. It is a useful confirmation that will be of interest to clinical researchers, as the observational studies are far from clear and there are not randomly controlled trials on the topic. 

The data is sensibly chosen and the analysis method is well done and sufficient to warrant their conclusion. The method is explained in sufficient detail in the main text - some of this could be summarized to make a more streamlined paper, and the detail moved to a supplementary section - and should be reproducible by anyone with access to the data. 

The paper is sensibly laid out and should be possible for a interested non-specialist to follow. 

No mention is made of the STROBE-MR guidelines, but the paper has followed all the major points. 

Figure 2 shows a convincing lack of any pattern, supporting the null result found; however it would be clearer to restrict to just the major method used in the paper (ivw, median, mode, mr-egger) and to add error bars to the points. 

I think the power calculations are slightly underestimated. The numbers given in the paper give me 80% power to detect an effect of size 1.23, and 87% power to detect 1.25 (from https://cnsgenomics.com/shiny/mRnd/) 

The discussion section needs to include a wider discussion of the current literature. This null result has been shown before in https://academic.oup.com/ije/article/48/5/1425/5569493 "Phenome-wide Mendelian-randomization study of genetically determined vitamin D on multiple health outcomes using the UK Biobank study" (This is easy to miss as the null results are discussed individually in the main body of the paper, but Type 1 diabetes is one of the outcomes in MR-PheWAS and it was no significant as show in this image https://academic.oup.com/view-large/figure/178161527/dyz182f2.tif ) . The instrument they used was weaker (2.84% of trait variance explained using 6 snps) and this confirmation of that result is useful. Clarification as to the degree of overlap in individuals in the outcome datasets & SNPs under consideration between the two studies would be helpful.

A greater prominence should also be given to a discussion of how differing geographic areas of studies may impact the SNPS results. "Variation in Associations between Allelic Variants of the Vitamin D Receptor Gene and Onset of Type 1 Diabetes Mellitus by Ambient Winter Ultraviolet Radiation Levels: A Meta-Regression Analysis" (https://academic.oup.com/aje/article/168/4/358/106033 ) showed that impact of SNPs on T1D varies with UV levels, suggesting that the impact in vitamin D deficient groups differs from that in non-deficient population. How homogeneous is the population under consideration in terms of geographic location? This could also be indicative of a non-linear effect. The authors say that traditional MR methods assume linearity, but there are a range of techniques adapting MR to non-linear situations - https://pubmed.ncbi.nlm.nih.gov/28317167/ or https://www.bmj.com/content/364/bmj.l1042 . If the authors have appropriate data to apply these, they should be considered. 

Under limitations, the authors should also explicitly discuss the lack of applicability to other ethnic groups, and potentially other geographic areas. 

*** Reviewer #2: 

Manousaki et al. used two-sample Mendelian Randomization analyses (MR) to investigate whether genetically determined lower vitamin D status is causally associated with higher risk of type 1 diabetes (T1D). While previous MR analyses of vitamin D status with other outcomes typically have used a 4 SNP genetic risk score as the instrument, the current analysis uses 69 SNPS, from a recent GWAS of around 440 000 participants in UK Biobank. However, these still only explains a small proportion of the variance of vitamin D status (circulating 25-hydroxyvitamin D, 25OHD). 

The main analysis showed that a one standard deviation decrease in standardized natural log-transformed 25OHD was not significantly associated with increased risk of T1D, OR=1.09, with moderately wide 95% CI of 0.86 - 1.40. Results seem reasonably robust to sensitivity analyses assessing potential influence of pleiotropy. The authors concluded that a large effect of vitamin D status on T1D is unlikely. In lightly of the suggestive findings from other studies and theoretical potential for initiating intervention studies to prevent T1D, the current study represent important evidence. Large data sets combined with analyses that seem competently executed is a strength. My comments are mostly minor. 

COMMENTS

1.Presentation of existing evidence from observational evidence: There is a relatively large literature linking various aspects of vitamin D with T1D, but my assessment is that the authors have not done a balanced assessment / selection of papers to discuss. I would suggest that the authors focus on available prospective studies of 25OHD in the current context. "Cross-sectional" studies comparing 25OHD in people diagnosed with T1D compared to controls are not high-quality epidemiological evidence, as these are clearly much more prone to both selection and reverse causation bias (e.g. refs 5-7), than are prospective studies (e.g. ref 9 and 12). One of the largest prospective studies in the field, with longitudinal 25OHD measurements from early childhood is the TEDDY study (Norris, Diabetes 2018). I would say this is a glaring omission from the intro and discussion. While TEDDY/Norris and TRIGR (Miettinen, ref 12) found evidence of an inverse association, the associations were moderate or weak, and not consistent with Simpson (ref 9) and Raab (ref 11). While islet autoimmunity (IA) is a surrogate endpoint for T1D, it is important to note that prospective studies of 25OHD before IA are less susceptible to reverse causation bias than other study types in the field (but note that Raab measured 25OHD after seroconversion, and is prone to potential reverse causation bias!). Yet another important, prospective study is Makinen/DIPP, J Clin Endocrinol Metab 2016, who found no significant association between longitudinal 25OHD and progression from IA to T1D. Again in the discussion (2nd paragraph), 3 low quality studies are cited to support observational evidence for a causal role of low 25OHD. I understand that it is nice if the story fits into the "MR narrative" where observational studies show an association and MR not, but a balanced review of existing high quality prospective studies are in my mind not consistently showing an inverse association with T1D. Finally, in one of the last paragraphs of the discussion, where the authors discuss limitations, ref 3 (Hypponen Lancet 2001) is cited to support that deviation from linear association between 25OHD and T1D should not be a problem. This is either a typo or an incorrect assessment of this paper. Hypponen 2001 did NOT include measures of 25OHD (and all of the association is based on less than 1% of the study population who deviated from the general advice to take vitamin D supplements), and clearly cannot be used to assess linearity of the potential 25OHD-T1D association. 

When the authors discuss the fact that their study cannot exclude the possibility that prenatal vitamin D status could influence T1D risk, I would suggest to cite two large scale prospective studies of maternal or neonatal 25OHD in relation to T1D (showing relatively precise estimates near null): Thorsen AJE 2018 and Jacobsen, Diabetologia 2016 (I am admittedly a co-author on one of them, so I may be biased, but you should at least assess these papers). 

Also reference to existing animal studies seems a bit unbalanced in my mind. Ref 8 cited in the introduction is not a study of mice as the authors state, but in vitro studies of human islets. There are many studies of various aspects of vitamin D and diabetes in NOD and other rodent models (in addition to ref 2), of various relevance to humans. 

2. SNPs and effects: I am not a GWAS or MR expert, but impression is that the analyses are competently performed. I have couple of comments which may reflect my of expertise. Given that most of the 69 SNPs were recently identified in UKBiobank, can we be confident that the R-square is not upwardly biased due to overfitting? I may be wrong, but after browsing the AJHG paper I could not see that the novel SNPs were replicated? I am worried that if the Rsq is overestimated, then so is the statistical power in the current analysis (and effect estimates probably also biased?). If the UK Biobank were the data set used to identify these novel SNPs, the effect estimates may have been biased (cf winners curse), and Rsq should perhaps have been estimated based on a replication dataset or at least cross validation (I may have browsed too quickly over this in the paper or cited reference, but these aspects should probably be better described). 

3. Population stratification? Again this question may reflect my lack of expertise in MR data analysis, but I wonder whether population stratification is properly adjusted for in the analyses. Even if the populations are of European descent, could not population stratification still confound results, if allele frequency and T1D risk differed systematically across strata? (or was this handled by adjusting each of the SNP-25OHD - and SNP - T1D associations for principal components?).

4. calculation and interpretation of statistical power

The paper cited for power calculations (Brien et al. 2013) is for continuous outcomes, not binary such as T1D. Would't it make more sense to use Burgess' method (IJE 2014), for binary outcomes? It may not make a major difference in practice, but would be good to clarify. 

Most importantly, based on the given power of 80% under an alternative hypothesis where the true OR is 1.25, the authors state in the end of the results section and again in the first paragraph of the discussion that "…results are robust to exclude large effects (OR >1.25), given ….". I do not think this is a proper interpretation. The test (or analysis) failed to detect a significant association, but did not exclude the possibility that the true OR is 1.25! 

MINOR COMMENTS

-a strength in the current study is that all 25OHDs were measured using a single assay (presumably in the same lab). However, the 12 cohorts used for T1D associations could perhaps be described in some more detail (age at onset, selection of controls, control of ethnicity and other types of population stratification, etc. (or did I miss this)

-The authors explained in the methods section that they presented the results in terms of a given contrast in 25OHD, which is important. It would probably help the readers if this was presented right away in the methods where this is explained (and perhaps briefly mentioned in the abstract too; it is after all not a major result). 

- While the authors have done a great job at assessing potential pleiotropy, and also discussing potential limitations of the study, my impression is that even with the "traditional" vitamin D pathway SNPs, we do not really know much about the functional consequences of the polymorphism (the fact that they are near a relevant enzymes is of course reassuring, but not enough). Furthermore, one of the loci most robustly associated with 25OHD is GC (encoding Vitamin D-binding protein, DBP), which in addition is also strongly associated with circulating DBP-concentrations. It is admittedly not well established, but a role of DBP in the aetiology of T1D has been suggested in a few studies. 

-The first reference cited for incidence of T1D in Canada, is incorrect (reference is about something else), probably a typo. Personally, I would cite a reference showing the incidence not only in a single country, and some of the variation across the world and over time. 

-Ref 19 is cited for expected risk in prevalence of T1D worldwide, but ref 19 is about something else! (genetic interactions)

-in the methods section, the authors state that they calculate risk associated with a SD increase, while the actual results were in fact presented as OR for an SD decrease.

-Ziegler ref 18 on skin pigmentation. Could this observation not simply have been due to ethnicity? I am not sure this is convincing evidence for reverse causation. 

*** Reviewer #3: 

This is a good mendelian randomisation study that tests the important hypothesis that higher vitamin D levels could protect from Type 1 diabetes. The authors have used the latest, recently greatly enlarged set of genetic variants robustly associated with vitamin D as a proxy for vitamin D levels, and the largest set of T1D case control data available (with genetics). In contrast to a study in 2011, with similar numbers of cases the authors find no evidence that Vitamin D levels alter the risk of type 1 diabetes. The methods are all very thorough and a wide range of appropriate sensitivity analyses performed

Main point. 

1. This might seem pedantic, but can the authors provide one more sensitivity analysis ? Using the 4 "canonical" SNPs rather than the 6 in the Jiang et al study. I ask for two reasons - the 6 SNPs in Jiang et included 2 outside of the key 4 that lie near key Vit D metabolism and synthesis genes, and it makes sense to exclude these from this very specific test. Second, the Cooper et al study in 2011, did see a positive association with these canonical variants, in what appears to be a very similar set of T1D cases. Third, fig 2 shows nicely how a very small number of much more specific-to-vitamin-D SNPs could have a big influence on the MR result. 

2. Many tests show considerable heterogeneity and some MR egger intercept p values suggest there could be meaningful levels of pleiotropy that are hard to properly account for. Many of these pleiotropic effects likely reflect the fact that most SNPs have strong effects on traits other than vitamin D. Although the methods used adjust the models for such effects, none can eliminate all the sources of error. Ultimately, the most useful advance for this question would be a larger sample size for the outcome trait - T1D cases and controls. Given that many of the MR results are trending towards a positive effect of Vit D in protecting from T1D, the question is crying out for a larger sample size.

Minor points:

1. Can a study be biased by confounding ? I think confounding is confounding ? likewise reverse causation is reverse causation , not bias ?

2. Suggest avoid chatty phrases such as "Widely known" and "on the other hand"

3. The background oversells MR a little - it is not immune from many of the issues you describe, such as reverse causation, confounding and not entirely equivalent to an RCT. Suggest qualify. 

Tim Frayling

***

[LINK]

---

## [Decision Letter · Decision Letter 2]

23 Nov 2020

Re: PMEDICINE-D-20-04125R2

"Decreased vitamin D levels and risk of type 1 diabetes: A Mendelian randomization study"

Dear Dr. Manousaki,

Thank you very much for submitting your revised paper, above, for consideration at PLOS Medicine. The revisions were seen again by all three reviewers, whose comments are enclosed below and at [LINK]; I hope you find them constructive. The reviewers feel that many of their original critiques have been responded to well although some issues remain in their reviews below; the reviewers ask for further modifications to clarify the results and present them in the context of prior evidence in a fuller way. 

Given the reviews we can't formally offer publication at this point but would ask you to revise the paper further in response to those reviews and the editors will then reassess. 

We expect to receive your revised manuscript by Dec 14 2020 11:59PM. Please email us (plosmedicine@plos.org) if you have any questions or concerns.

We look forward to receiving your revised manuscript. 

Sincerely,

Emma Veitch, PhD

PLOS Medicine

On behalf of Clare Stone, PhD, Acting Chief Editor, 

PLOS Medicine

plosmedicine.org

Format requests from the editors:

*The editors felt it would be good to also mention (briefly) additional possible limitations in the abstract (as are noted in the full discussion section, eg the possibility for pleiotropy and canalization, as well as the point already noted that very small effects can't be excluded.

*Many thanks for including the information about the lack of a prespecified analysis plan per journal policy - however, the point that "group has a long track of published MR studies" could probably be excluded.

*The funding statement in the main manuscript text can be deleted, as this is provided in the submission form (and needs only to be there, not in the actual manuscript as well). 

Comments from the reviewers:

Reviewer #1: I disagree with the author's choice not to discuss Meng et. al. paper because the "diseases/traits with null outcomes among the 920 tested are not outlined in the main paper or the supplement". It is clear from the paper (and supplement) that they have used the mapping from ( https://phewascatalog.org/phecodes ) which includes Type 1 diabetes, this can also be seen in reference 8 of the main paper ( https://academic.oup.com/bioinformatics/article/30/16/2375/2748157 ). Given the concern over exact case numbers, they could contact the authors. However the UK Biobank website shows it would be ~ 1000 people ( https://biobank.ctsu.ox.ac.uk/crystal/field.cgi?id=41202 ). 

This paper is a clear improvement on that study, as it is focused exclusively on type 1 diabetes and thus able to discuss and investigate the null result in detail, but I still feel adding the confirmation of this result in other sources is important - particularly when they use it to respond to reviewer 3's comments! 

Otherwise I think this paper has responded well to my comments and those of the other reviewers, and is ready to publish. 

Reviewer #2: The authors have responded and revised well, with one minor exception. I am not entirely happy with the interpretation of the power calculation. My main point was not about the aboslute power (now 80% power to detect OR of 1.23), but how such a power analysis is interpreted. The authors still say the study "excluded ORs larger than" the calculated smallest OR with 80%. I do not think this is formally correct. If anything, it may be reasonable to say that the study (reasonably) excluded ORs outside the 95% CI. THe fact that you had 80% power to detect true ORs of 1.23 or larger, and you failed, does not mean you excluded such values. At least this is how I have understood this. I am willing to consider a counterargument, but the authors did not really address this in their response. 

Reviewer #3: Many thanks for doing that extra 4 SNP analysis. It is slightly worrying how this result differs from the 2011 result, given the very similar set of outcome cases and SNPs - you might expand a little on why you think the result differs when using the same SNPs and v similar outcome data. I realise they didnt do 2 sample MR so did not accoutn for the SNP-exposure dosage - and I see that the strongest Vitamin D SNP is flat in your data and pulling any protective effect towards the null, but is this one that was associated in the Cooper et al study and what are the differences ?

[LINK]

---

## [Decision Letter · Decision Letter 3]

18 Dec 2020

Dear Dr. Manousaki,

Thank you very much for re-submitting your manuscript "Decreased vitamin D levels and risk of type 1 diabetes: A Mendelian randomization study" (PMEDICINE-D-20-04125R3) for consideration at PLOS Medicine.

I have discussed the paper with editorial colleagues and it was also seen again by one reviewer. I am pleased to tell you that, provided the remaining editorial and production issues are dealt with, we expect to be able to accept the paper for publication in the journal.

[LINK]

We hope to receive your revised manuscript after the holidays. Please email us (plosmedicine@plos.org) if you have any questions or concerns.

Please let me know if you have any questions. Otherwise, we look forward to receiving the revised manuscript soon.   

Sincerely,

Richard Turner PhD

rturner@plos.org

Requests from Editors:

Please submit both clean and tracked files for your next revision. 

To your competing interest statement, please add "GDS is a member of PLOS Medicine's Editorial Board." or similar. 

We suggest removing the word "Decreased" from the title. 

At line 54, should the numbers of cases and controls not add up to the total?

At line 60, please adapt the text as follows: “MR analyses suggested that a one standard deviation decrease in standardized natural log-transformed 25OHD (corresponding to a 29nmol/l change in 25OHD levels in vitamin D insufficient individuals) was not associated with an increase in type 1 diabetes risk (inverse-variance weighted MR OR=1.09, 95% CI: 0.86-1.40, p=0.48).”.

In the abstract and elsewhere in the paper, please avoid statements like "to reasonable/confidently exclude". We would suggest language such as "Our findings indicate that decreased vitamin D levels did not have a substantial impact on risk of type 1 diabetes in the populations studied. Study limitations include an inability to exclude the existence of smaller associations, and a lack of evidence from non-European populations.".

At line 81, please make that "are accurate".

At line 82, please add "to our knowledge".

At line 89, please make that "an alternative".

At line 92, please revise the bullet point to: "Our study did not find evidence in support of a large effect of vitamin D levels on type 1 diabetes. However, the findings do not exclude the possibility that there may be smaller effects than we could detect.” or similar.

At line 98, please revise the bullet point to: “Our results do not support increasing vitamin D levels as a strategy to decrease the risk of type 1 diabetes.”

Please remove the word "Specifically" at lines 110 and 334.

At line 166, please make that "to be".

Around line 200, please adapt the language to "We provide a completed STrengthening the Reporting of OBservational studies in Epidemiology (STROBE) checklist for the study (See S1_STROBE_Checklist)." or similar, and rename the relevant attachment to match. 

At line 391, please remove the word "simply" (you may wish to adapt the text to "straightforward associations ..."). 

At line 411, please substitute "estimate", or similar, for "test".

At line 465, please reword "failed to identify", e.g., to "... identified no large impact ...".

Throughout the paper, please remove spaces from within the reference call-outs (e.g., "... diabetes [12,13].").

Please revisit table 1. We generally ask that p values are quoted as "p<0.001" where appropriate, unless there is a specific statistical reason to do otherwise. There is also a value with two decimal points in this table. Please correct all misspellings of "intercept".

Please also look over the supplementary tables, correcting any similar issues. 

Please reformat the attached STROBE checklist, which would be much clearer if the entries regarding locations of specific items in the present paper were organized into a right-hand column. 

Comments from Reviewers:

*** Reviewer #3: 

thanks for the extra info about the 4 SNP instrument. Sorry to prolong the discussion, but i think the fact that you have estimated the exposure effects in 430k people rather than 2k people could be an explanation for the different results, rather than the stronger instrument - you say that you explain more variance in vitamin D but you have also added in a lot more pleiotropy - and pleiotropy that might not be properly accounted for by MR egger - especially if the INSIDE assumptions are violated - which is possible if there is a consistent dose response effect of lipid SNP > vitamin D levels as well as SNP > lipid levels. Can you say how much more variance you explain compared to the 4 canonical SNPs in your added discussion paragraph , and point out the that Egger is not a perfect solution to pleiotropy ?

***

[LINK]

---

## [Editor Report · Decision Letter 4]

12 Jan 2021

Dear Dr Manousaki, 

On behalf of my colleagues and the Academic Editor, Prof Frayling, I am pleased to inform you that we agree to publish your manuscript "Vitamin D levels and risk of type 1 diabetes: A Mendelian randomization study" (PMEDICINE-D-20-04125R4) in PLOS Medicine.

PRESS

Sincerely, 

Richard Turner, PhD 

rturner@plos.org